# Meta-Analysis of 16S rRNA Sequencing Reveals Altered Fecal but Not Vaginal Microbial Composition and Function in Women with Endometriosis

**DOI:** 10.3390/medicina61050888

**Published:** 2025-05-14

**Authors:** Astrid Torraco, Sara Di Nicolantonio, Martina Cardisciani, Eleonora Ortu, Davide Pietropaoli, Serena Altamura, Rita Del Pinto

**Affiliations:** 1Department of Clinical Medicine, Public Health, Life and Environmental Sciences, University of L’Aquila, 67100 L’Aquila, Italy; astrid.torraco@student.univaq.it (A.T.); sara.dinicolantonio@graduate.univaq.it (S.D.N.); martina.cardisciani@student.univaq.it (M.C.); eleonora.ortu@univaq.it (E.O.); 2Department of Physical and Chemical Science, University of L’Aquila, 67100 L’Aquila, Italy

**Keywords:** endometriosis, oral–gut microbiota, dysbiosis, vaginal microbiota, meta-analysis

## Abstract

*Background and Objectives*: Dysbiosis of the oral–gut axis is related to several extraintestinal inflammatory diseases, including endometriosis. This study aims to assess the microbial landscape and pathogenic potential of distinct biological niches during endometriosis. *Materials and Methods*: A microbiome meta-analysis was conducted on 182 metagenomic sequences (79 of fecal and 103 of vaginal origin) from women with and without endometriosis. Fecal and vaginal microbial diversity, differential abundance, and functional analysis based on disease status were assessed. Random forest, gradient boosting, and generalized linear modeling were used to predict endometriosis based on differentially enriched bacteria. *Results*: Only intestinal microbes displayed distinctive taxonomic and functional characteristics in women with endometriosis compared to control women. Taxonomic differences were quantified using the microbial endometriosis index (MEI), which effectively distinguished between individuals with and without the disease. The observed functional enrichment pointed to proinflammatory pathways previously related to endometriosis development. *Conclusions*: Dysbiosis in the oral–gut microbial community appears to play a prevalent role in endometriosis. Our findings pave the ground for future studies exploring the potential mechanistic involvement of the oral–gut axis in disease pathogenesis.

## 1. Introduction

Endometriosis is an estrogen-dependent gynecological condition affecting women of childbearing age, characterized by the presence of endometrial tissue outside the uterine cavity, causing pelvic pain and infertility. The prevalence is estimated to be around 5% between the ages of 25 and 35 with an annual incidence of 0.1% in women between the ages of 15 and 49. However, accurate prevalence data are jeopardized due to the need for surgical visualization to confirm the diagnosis [1].

The pathogenesis of endometriosis remains elusive: in the past few years, the most credited hypothesis was that of ‘retrograde menstruation’, which assumed endometrial tissue to move into the pelvic cavity due to non-synergic uterine contractions [1]. However, such contractions are not considered a necessary and sufficient condition for developing the disease.

In recent times, alongside genetic factors involving estrogen-regulated pathways, female sex development, and intercellular adhesion [2], a growing number of studies indicate an association between pelvic inflammatory disease (PID), caused by the ascension of pathogenic bacteria from the vagina, and endometriosis [3,4]. Such a trigger could promote the secretion of inflammatory cytokines and chemokines, thereby facilitating angiogenesis and the ectopic implantation of endometrial tissue aggregates [3,4]. One last pathogenetic hypothesis involves dysbiosis of the oral–gastrointestinal tract. An alteration in the oral and gut microbiota composition and functioning is associated with various diseases [5,6,7], and recent evidence points to the presence of perturbations in the oral and gut microbial composition also during endometriosis [8,9,10]. Of note, most women with endometriosis may also experience intestinal symptoms alongside gynecological symptoms. Gastrointestinal symptoms such as abdominal pain, bloating, constipation, diarrhea, and painful bowel movements have been frequently reported [11,12]. While some symptoms are linked to the mechanical effects of lesions (e.g., bowel obstruction) [13], others, including cyclic bloating and IBS-like symptoms, are attributed to immune-mediated inflammation and altered prostaglandin release even without direct intestinal involvement [14].

Furthermore, increased odds of periodontitis were found in women with endometriosis [15], suggesting the hypothesis of a generalized, global immune dysregulation as a potential underlying link between the two conditions, which is possibly promoted by oral dysbiosis. However, the link between oral–gut dysbiosis and endometriosis remains largely unexplored. In this regard, a meta-analytic approach could offer a novel opportunity to capture the microbial landscape and pathogenic potential of distinct biological niches during endometriosis.

In this study, by pooling individual patient data in a microbiome meta-analysis of gut and vaginal metagenomes, we tested whether any divergence exists in the composition and functional profiles of the fecal and vaginal microbial communities in women living with endometriosis compared to healthy women and whether such differences could discriminate between health and disease status. By identifying the distinctive taxonomic and functional characteristics limited to the intestinal environment in patients with endometriosis compared to control women, our findings pave the ground for future studies exploring the potential mechanistic involvement of the oral–gut axis in disease pathogenesis.

## 2. Materials and Methods

### 2.1. Collection of Datasets

The NCBI bioproject database was used to locate studies within the International Nucleotide Sequence Database Collaboration (INSDC) network (https://www.insdc.org/about-insdc/, accessed on 12 December 2021) that published gut metagenomic data from women with endometriosis and healthy female controls. A search was performed for the following MeSH terms: Microbiota [MeSH Major Topic] AND Endometriosis [MeSH Major Topic], Microbiota [MeSH Major Topic] and Genital Diseases, Female [MeSH Major Topic].

Observational or intervention studies with publicly available fecal and/or vaginal microbial 16S RNA sequencing data (raw FASTQ files) and metadata indicating case (endometriosis) or control (healthy) status were considered admissible.

Non-English studies, review articles, nonhuman studies, case reports, studies on specific conditions/diseases different from endometriosis, studies reporting on explicit antibiotic/probiotic/prebiotic use, non-metagenomic or archaeological studies, and studies without metadata of interest were excluded. Studies requiring further ethics committee approval, access permissions (e.g., controlled database of Genotypes and Phenotypes [dbGaP] studies) or personal communication with authors were also excluded. No restrictions were applied based on geographic area, race/ethnicity, or endometriotic disease activity. When available, demographic identity, age, BMI, and disease stage were collected as reported in the original studies.

Four studies [16,17,18,19], including 389 metagenomic sequences, were screened (Appendix A), and project titles, descriptions, and other information were exported to a spreadsheet for subsequent manual eligibility checks.

As of January 2024, 182 metagenomic sequences from two bioprojects (accession numbers: PRJNA722289 and PRJNA424567) [18,19] met the inclusion criteria (Appendix A).

All the included metagenomic sequences had been processed and analyzed using the Illumina MiSeq platform, while DNA extraction occurred with the RNeasy Mini Kit (Qiagen, Hilden, Germany) for BioProject PRJNA722289 and with the MoBio Powersoil extraction kit (MoBio Laboratories Inc., Carlsbad, CA, USA) for BioProject PRJNA424567.

Both the case and control groups were extracted from these two bioprojects.

Individuals labeled as “healthy” or “control” by the original authors, and who had no record of gynecologic or other exclusionary diseases, were selected as controls. The inclusion criteria for the control group were (i) the availability of raw 16S sequencing data, (ii) the presence of relevant metadata, and (iii) designation as “control” in the original dataset. The control group (N = 74) was frequency-matched to the case group (N = 108) by sample type (fecal or vaginal) and age range where possible as available in the original metadata.

Of the 182 samples analyzed, 79 (43.4%) were of fecal origin, and 103 (56.6%) were vaginal [18,19]. Therefore, two separate cohorts of individuals with endometriosis (N = 108; 44.4% with fecal microbiota data and 55.6% with vaginal microbiota data) and healthy individuals (N = 74; 41.9% with fecal microbiota data and 58.1% with vaginal microbiota data) were generated. Bioinformatics analysis was implemented in R (version 4.2.2).

### 2.2. 16S rRNA Preprocessing

Raw data were downloaded and processed through our 16S rRNA processing pipeline. Amplicon resolution (ASV) was performed using the DADA2 algorithm. Specifically, FASTQs were truncated according to primer length as specified in the original article when available; otherwise, trimming was performed at 20 bp for F/R ends. After visual inspection, sequences were qualitatively filtered by truncating to the first base with a quality score Q < 25. Amplicon sequence variants (ASVs) were then assigned to taxonomy using the SILVA database version 138.1 [20]. For each dataset, samples with fewer than 100 reads (0 samples removed), ASVs with fewer than 10 reads, and ASVs present in less than 1% of samples within a study were removed. The relative abundance of each ASV was calculated by dividing its value by the total number of reads per sample using a specific R library. The taxonomy table, ASVs, and metadata were condensed into a single phyloseq object and used for further analysis.

### 2.3. Microbiome Community Analysis

Based on relative abundances, alpha diversity was calculated as the observed-alpha diversity index [21], and beta diversity was calculated using the Bray–Curtis similarity index [22]. The mean abundance of each phylum or genus was calculated as the average of the respective mean values. Patients with zero abundance were excluded from the calculation of genus abundance.

### 2.4. Analysis of Confounding Factors

Although the two cohorts were balanced between healthy and endometriosis patients by age and sampling sites (gut, vagina), other approaches were also adopted to control for major confounders. The effect of potential confounders on individual microbial characteristics was quantified, and ANOVA was performed on the rank-transformed data [23]. The rank transformation was performed to account for the non-Gaussian distribution of the microbiome abundance data. Using a generalized linear model (GLM), the Log10 association phyla abundance was normalized with selected metadata (age, disease status, sampling site, study, and library size). Age was used as a continuous variable. The variable “study” encompasses any possible methodological variability between bioprojects. Then, the total variance within the individual microbial phyla was compared to the variance explained by possible confounding factors (i.e., study and sampling site) in analogy to a linear model, where putative confounding factors were included as explanatory variables for phyla abundance. Stratified analyses based on sampling sites and disease status were also performed.

### 2.5. The Microbial Endometriosis Index (MEI)

Based on the previous literature [24,25], we calculated the microbial endometriosis index (MEI) as a summary statistic of the disease-specific fecal microbial diversity. To this aim, we first derived the relative abundance of fecal genera using log2 normalization. Fecal genera enrichment in women with endometriosis and healthy control women was assessed by applying the FDR to a *t*-test [24], and the index was calculated as the logarithm of the ratio between the relative abundance sum of disease-enriched genera +1 and the relative abundance sum of the healthy-enriched genera +1. Relative abundances were transformed into account for datasets containing zeroes, so that the MEI ranged from negative values (indicating health-specific richness) to positive values (indicating disease-specific richness). A MEI of zero indicated similar taxonomic richness in health and disease.

### 2.6. Statistics

#### 2.6.1. Taxonomic Analysis

Differential abundance analysis (DA) was used to identify site-specific microbiome biomarkers in healthy women and patients with endometriosis. Since data visualization at multiple levels of the taxonomic hierarchy provides insights into broad patterns and finer details within the microbiome [26], analyses were performed at different taxonomic levels. Specifically, ASVs were agglomerated into phyla or genera, reflecting major microbial groups or more detailed microbial communities, respectively, and three different approaches, based on different DA methods, were applied. They were then combined to increase the robustness of the results and provide more context to improve their biological interpretation, according to the literature [27,28]. Specifically, Welch’s *t*-test [26], differential expression analysis for sequence count data version 2 (DeSeq2) [29], and Linear discriminant analysis Effect Size (LEfSe) [30] were implemented using specific R libraries in the Microbiome package [31,32].

For DA testing, we created a custom R script file, implemented primarily with the MicrobiomeMarker library [32], which enables the creation of a unified set of tools for discovering microbiome biomarkers by integrating existing widely used DA methods [32]. The analyses were performed on the phyloseq object described previously. Significance was set to an alpha value of 0.05, and the FDR *p*-values were adjusted using Benjamini–Hochberg correction as needed [33]. Based on these specifications, if at least two methods agreed, microorganisms were considered differentially abundant in health versus disease status. Three approaches were used to perform the supervised classification of health and disease states based on distinct fecal microbial characteristics with and without the inclusion of age as a covariate: random forest (RF), generalized linear model (GLM), and gradient boosting model (GBM). Machine learning classifiers were trained using the caret R package (v6.0-94). Ten-fold cross-validation repeated three times was used for model validation (trainControl (method = “repeatedcv”, number = 10, repeats = 3)). GBM models were optimized over shrinkage values of 0.1 and interaction depths of 1–5; the final model had 100 trees and depth = 3. RF models were trained with 500 trees with mtry optimized within cross-validation. Logistic regression models used MEI and other features without regularization. All models were evaluated using the MLeval package to compute ROC curves, AUC, and classification metrics.

#### 2.6.2. Functional Microbiome Analysis

Following taxonomic composition analysis, functional microbiome analysis was conducted on gut microbial communities of women with and without endometriosis. To this end, PICRUSt2 (v2.6.1) (Phylogenetic Investigation of Communities by Reconstruction of Unobserved States) was applied. Briefly, PICRUSt2 [34] is a bioinformatics software package that predicts metagenome functional profiles and phenotypes from 16S rRNA gene sequences using a reference phylogeny and a large database of reference genomes and gene families [34]. Ggpicrust2 library for R [35] was used for DA analyses and for generating relative data visualizations to highlight functional signals from the PICRUSt2 output. Precisely, we calculated the relative abundance of functional gut microbiome pathways comparing women with and without endometriosis based on DESeq2 [29]. Between-group differences derived from DA analysis were expressed as Log2 fold changes with FDR *p*-values adjusted using Benjamini–Hochberg correction [33] with a default cutoff value of 0.05. DESeq2 was selected as the primary tool for functional inference due to its robustness in controlling type I error in sparse and overdispersed metagenomic data. Compared to LEfSe [30] and t-test-based methods [26], DESeq2 applies variance-stabilizing transformations and shrinkage estimators that enhance reproducibility particularly when working with uneven sampling depth across public datasets [36,37].

### 2.7. Study Approval

As an analysis of existing, deidentified data, IRB approval was deemed exempt.

### 2.8. Data Availability

Raw sequencing data for each study can be accessed as described earlier. Raw processed ASV tables can be accessed on GitHub (v0.1) and available at https://github.com/PietropaoliLab/ (accessed on 10 May 2025). All other relevant data supporting the findings of this study are available in the present manuscript and its supplementary data or upon request to the corresponding author. This meta-analysis was not registered in any database.

## 3. Results

### 3.1. Data Features Included in the Meta-Analysis

In this meta-analysis of fecal and vaginal microbiomes, there were a total of 182 metagenomic sequences (N. 108 from women with endometriosis; N. 74 from healthy control women), two of which were included from BioProjects (PRJNA722289; PRJNA424567) with available 16S amplicon sequencing raw data (FASTQ or FASTA) from gut and vaginal samples of women with endometriosis and healthy control women as well as metadata on disease status. The metagenomic sequences assessed the V3–V4 regions. We then derived two cohorts of participants based on their health condition (women with endometriosis and healthy control women). Women within each condition did not differ in age, BMI, menstrual cycle phase, and sampling sites (Table 1).

A principal component analysis (PCA) was conducted to determine which variables contributed the most to the variance in the data analyzed (Figure 1). PCA revealed that technical factors, including sampling site and BioProject, explained the largest portion of inter-sample variability (Figure 1A,C). These findings underscore the importance of addressing batch effects in meta-analyses of microbiome data. However, the disease group (endometriosis vs. control) showed greater contribution to Dim2, which was orthogonal to the primary technical axis (Dim1), suggesting that the disease-associated microbial shifts are not fully confounded by batch or site effects. To further mitigate this, all downstream analyses were stratified by biome (rectal vs. vaginal), ensuring that biological inferences reflect within-biome differences rather than cross-biome or inter-cohort variability.

### 3.2. Microbial Diversity Between and Within Samples

Figure 2A shows the alpha diversity calculated between the examined samples. Interestingly, significant differences are present between the fecal microbiome of women with endometriosis and healthy control women, keeping with the hypothesis of a possible role for the oral–gut axis in the pathogenesis of endometriosis. By contrast, the vaginal microbiome did not differ by health status. The analysis of beta diversity indicates greater richness in the vaginal than in the fecal microbiome composition independent of disease status (Figure 2B). We then explored whether participants’ age had an impact on the fecal or vaginal microbiome diversity. Fecal and vaginal microbiome diversities were found to have no correlations with age in women with and without endometriosis (Figure 2C). We, therefore, concluded that the observed alpha diversity in the fecal microbiome composition of women with endometriosis is not determined by age, possibly reflecting disease-related perturbations in local microbiota.

### 3.3. Taxonomic Relative Abundance in Fecal and Vaginal Samples

In the vaginal samples, the phylum Euglenozoa was more abundant relative to other phyla in the endometriosis group, while Actinobacteriota, Bacteroidota, and Proteobacteria were more abundant in the control group (Appendix A). In terms of genera, Streptococcus and Lactobacillus were more abundant than other genera in women with endometriosis, while Finegoldia and Prevotella were more abundant in control women. As for the fecal microbiota, Actinobacteriota were more abundant relative to other phyla in the endometriosis group, while Proteobacteria were more abundant in the control group (Appendix A). In terms of genera, Anaerococcus, Fenollaria, Lactobacillus, and Peptoniphilus were more abundant relative to other bacteria in women with endometriosis. At the same time, Acinetobacter, Finegoldia, Streptococcus, and UCG-002 were more abundant in healthy control women (Appendix A).

### 3.4. Differentially Abundant Bacteria and Disease Status Prediction

Finally, we wanted to determine which vaginal and fecal bacteria were differentially abundant in women with endometriosis as compared to healthy control women. To this aim, a combination of methods was applied to increase reliability and consistency and improve interpretation of findings, according to literature. Based on the consensus level between methodologies, our results indicate that 14 fecal genera are differentially abundant in women with and without endometriosis (Figure 2D). Specifically, *Alloprevotella*, *Succinivibrio*, and Rikenellaceae RC9 gut group were found to be enriched in the fecal samples of healthy women, while *Facklamia* and *Corynebacterium* were enriched in the gut of women with endometriosis (Figure 2E). Conversely, no genera were found to be differentially abundant in vaginal samples of women with and without endometriosis, further suggesting that gut, and not vaginal, microbiome perturbations could be relevant during disease development (Figure 2D).

Consistent with this finding, the differentially abundant fecal bacteria could accurately predict the disease status, with increased predictive accuracy when age was also considered (AUC-ROC for RF: 0.85; AUC-ROC for GLM: 0.82) (Figure 2F).

### 3.5. The MEI as a Measure of Disease-Specific Diversity

We then asked whether the differential enrichment observed during endometriosis and predictive of disease status could be summarized in a quantitative measure of disease-specific diversity. To this aim, we calculated a summary statistic of the observed microbial diversity in health and disease states, with increased enrichment for values different from zero. The disease-specific enrichment observed in fecal samples of women with endometriosis was expressed by a positive MEI, which was in contrast with the negative MEI of fecal samples from healthy control women (*p* = 0.0014) (Figure 2G). In keeping with the observed increased predictive power of models when both bacteria enriched in endometriosis and age were considered, an inverse correlation was found between age and MEI (Appendix A).

### 3.6. Results of Functional Microbiome Analysis

Following the taxonomic analysis and the finding of a disease-specific diversity only in the gut (and not the vaginal) microbiome composition, we aimed to assess any difference in the functional potential of gut microorganisms based on participants’ health condition. Several functional pathways were differentially enriched in women with and without endometriosis (Figure 3). In women with endometriosis, we found an enrichment in functional pathways related to proteasome, cellular adhesion and communication, metabolism, secretory digestive functions, and, interestingly, cardiomyopathy. Healthy control women showed an enrichment in functional pathways related to transcription and translation, glycolipid metabolism, and cellular invasion.

## 4. Discussion

The results of the present meta-analysis of the fecal and vaginal metagenomes of patients with and without endometriosis reveal that only the former exhibits distinctive characteristics in women with endometriosis compared with control subjects and can predict the status of the disease correctly. This observation was paralleled by the finding of enrichment in functional pathways in the gut that were either causally related to endometriosis (proteasome) or associated with other inflammatory conditions (cardiovascular diseases). Taken together, these findings suggest that gut, and not vaginal, microbiota perturbations could be relevant during disease development by potentially disrupting immune homeostasis, thereby carrying potential diagnostic implications.

Previous evidence on the comparison of vaginal and gut microbiome in women with and without endometriosis indicates overall similarity in their composition and subtle differences in the relative abundance of some genera with a dominance of Escherichia and Shigella in feces and absence of the genus Atopobium in the vagina [16]. Of note, the latter finding is in contrast with reports from other studies [38]. Differential abundance analysis was also not performed in the study [16]. Conversely, our observation of increased relative abundance of the genus Streptococcus in vaginal samples of women with endometriosis aligns with the findings of recent systematic and narrative reviews [38,39]. It is interesting to note that several proinflammatory cytokines, including NF-κB, IL-1, IL-8 and cyclooxygenase-2 (COX-2), can be inducibly overexpressed by Streptococci [40,41], suggesting a connection between dysbiosis and the proinflammatory environment that characterizes endometriosis. In parallel, an analysis of the microbial composition of fecal samples and cervical mucus of women with and without endometriosis indicated that endometriosis patients harbor distinct microbial communities versus control group especially in feces, where the depletion of protective microbes was observed, and that the gut microbiota was superior to cervical microbiota in classifying endometriosis, which is in line with our findings [17]. Unfortunately, due to lack of metadata on health/disease status, the above-mentioned studies [16,17] could not be included in our microbiome meta-analysis. The observed functional enrichment of fecal bacteria, pointing to proinflammatory pathways previously related to endometriosis development, coupled with the clinical evidence of common peritoneal location of the ectopic endometrium during the disease [17], support a biologically plausible prevalent role for the oral–gut microbial community in the disease. However, a limitation for the accuracy of predictive models in our and others’ studies [17] may be the handful of samples included. In keeping with this, we cannot rule out a role for unmeasured confounders limiting the predictive value of vaginal metagenomes, a consideration that further supports the urge for further publicly available research in the field.

The gut microbiome has always been considered the body’s largest resident microbial habitat. It holds a crucial role in several physiological processes, including digestion/fermentation and metabolism, vitamins and hormones synthesis, protection against pathogens, and innate immunity maturation and homeostasis [42]. Consistent evidence indicates that alterations in the composition and function of the gut microbiome appear to play an essential role in the pathogenesis of several diseases [28]. Differences in the diversity and composition of the fecal microbial community have been related to diseases of the oral–gastrointestinal tract as well as inflammatory diseases affecting other systems, spanning Crohn’s disease [43] and ulcerative colitis to cardiovascular, metabolic, and neurological diseases [44]. Our functional analysis also points to the connection between fecal microbial perturbations and inflammation. In fact, we found enriched functional pathways related to proteasomes in the feces of endometriosis women. Proteasomes are key regulators of cellular homeostasis [45], and dysregulation in their expression and functioning may reduce sensitivity to apoptosis [46] and induce inflammation by the constitutive activation of NF-kB [45], thereby potentially contributing to the development of ectopic endometrial lesions. In keeping with this, treatment with a proteasome inhibitor was found to significantly reduce the burden of experimentally induced endometriosis in rats [47]. In addition, our finding of an enrichment in functional pathways related to cardiovascular diseases in feces of women with endometriosis is particularly interesting, as the dysbiotic/dysfunctional oral microbiome has been consistently associated with chronic inflammatory diseases, such as atherosclerosis and cardiovascular diseases, by directly or indirectly stimulating immune dysregulation [10]. Additionally, recent studies have reported an increased risk of heart failure among women with endometriosis, which is a condition that often represents the clinical manifestation of various forms of cardiomyopathy [48]. Mechanisms such as systemic inflammation, oxidative stress, endothelial dysfunction, and imbalances in estrogenic hormonal regulation are likely to underlie this association [49]. In this context, the identification of pathways related to cardiomyopathy in our study may reflect the presence of shared pathogenic processes between endometriosis and cardiovascular diseases.

As for endometriosis and oral–gut dysbiosis, previous studies support the idea of a bidirectional relationship between the two [50,51]. Various evidence suggests that proinflammatory factors and immune system perturbations are central to the pathogenesis of endometriosis [52]. Additionally, intestinal dysbiosis may alter the metabolome by increasing the production of metabolites in the brain that stimulate neuronal receptors, including GnRH receptors. This, in turn, leads to elevated circulating estrogen levels, which are implicated in the development of endometriotic lesions [53,54]. Intestinal dysbiosis can affect beta-glucuronidase production, which is an enzyme typically secreted in the gut by Bacteroides, Bifidobacterium, Escherichia coli, and Lactobacillus. This leads to increased levels of circulating active estrogen, which has been linked to the onset of cancer, polycystic ovary syndrome (PCOS), fertility problems, and, indeed, endometriosis [55]. Notably, a 2023 study proposed gut microbial beta-glucuronidase (gmGUS) as a biomarker for the early diagnosis of estrogen-dependent diseases [56,57]. In our analysis, Lactobacillus was found to be relatively more abundant than other genera in both vaginal and fecal samples of women with endometriosis. Indeed, a high-estrogen environment could facilitate the increased metabolic activity of lactic acid–producers, such as Lactobacilli [58]. An increase in Lactobacilli could also indicate an attempt by the body to maintain a microbial balance and counteract the inflammatory process during the disease. Indeed, Lactobacillus species, widely present from the oral cavity to the lower gastrointestinal tract, exert diverse and even opposite functions regarding microbial balance, immune processes regulation, and disease development [59].

Within the continuum of the oral–gut environment (the so-called “oral–gut axis”), oral bacteria can participate in the pathogenesis of endometriosis by several mechanisms, including translocation to the gastrointestinal tract, causing dysbiosis and inflammation and infection by bloodstream dissemination. Indeed, immunohistochemical and biochemical analyses on women with endometriosis revealed that Fusobacterium, a common member of the oral and gastrointestinal tract microbiota that can cause periodontal disease, was enriched in their endometrium and endometrial lesions compared to control samples [9]. Fusobacterium was also found to be enriched within the oral samples of women with moderate/severe endometriosis [8], which is consistent with the finding that periodontitis is more common in endometriosis than in healthy women [15]. Moreover, the infection of endometrial cells by Fusobacterium in a mouse model facilitated endometriosis lesion development by leading to the transition from quiescent fibroblasts to active myofibroblasts, and antibiotic therapy reverted such burden [9]. With its pathogenic traits like adhesion and invasion, Fusobacterium has been increasingly associated with several diseases involving the inflammatory processes [60,61]. Although we did not observe an enrichment in Fusobacterium, it is interesting to note that we found a functional enrichment in cellular adhesion pathways in fecal samples of women with endometriosis.

In our study, the genera *Corynebacterium* and *Facklamia* were uniquely enriched in the gut of women with endometriosis. While the virulence of *Facklamia* has not been fully determined, and it was hypothesized to be part of normal flora in the female genital tract, it was also reported to be implicated in invasive infections [62]. Corynebacteria are opportunistic pathogens capable of causing infections in immune-compromised individuals or if they penetrate sterile sites [63]. In the oral cavity, *Corynebacterium* spp. is known to form the structured habitat where other taxa can locate; at the level of the genital tract, *Corynebacterium* was associated with men-specific infections [64] and was also found to be highly abundant in women living with endometriosis [65].

The hypothesis that an alteration in the oral–gastrointestinal microbiota may be related to endometriosis opens the door to potential diagnostic, preventive, and therapeutic possibilities. Probiotics, particularly Lactobacillus, balance immunological and inflammatory responses, secreting antibacterial compounds and restoring the oro-gastrointestinal microbiota [66]. Studies in mice have also found that treating endometriosis with Lactobacillus improves endometriosis symptoms and reduces lesion growth [67]. This effect is thought to be related to the ability of probiotics to reverse endometriosis immune dysregulation with an increase in IL-6, IL-12, and NK cells [68]. In a pilot study in women with endometriosis, the oral administration of Lactobacillus had beneficial effects on chronic pelvic pain and dysmenorrhea after eight weeks of therapy [69]. A similar positive effect was also found in women with PCOS with reduced levels of inflammatory markers such as IL-10, TNF-α, and CRP [70]. However, it must be noted that current evidence does not support the therapeutic application of prebiotics and probiotics in the disease.

This study is not without limitations. The meta-analysis does not consider all studies evaluating the rectal and vaginal microbiota in endometriosis and non-endometriosis patients, mainly due to the unavailability of microbiota data and metadata from public archives, which might limit the generalizability of our findings. Furthermore, the absence of metadata on potential variables of interest, such as disease duration, complications, smoking status, dietary habits, and antibiotic use, or their limited availability as to disease stage and disease severity, prevented any additional specific or comprehensive subgroup analysis. While the exclusion of datasets lacking metadata (e.g., disease duration, antibiotic exposure, gastrointestinal symptomatology) may influence the generalizability of our results, such limitations are common in large-scale microbiome meta-analyses. Prior studies have demonstrated that antibiotic usage and chronic inflammation significantly modulate gut microbial composition [71]. Moreover, the lack of detailed clinical information, such as gastrointestinal symptom profiles, limited the possibility of correlating microbial alterations with symptomatology directly related to the hypothesized pathophysiological mechanisms. Metadata completeness remains inconsistent across repositories, constraining stratified subgroup analysis. Future prospective studies with harmonized clinical and environmental metadata are warranted to validate and expand these findings.

## 5. Conclusions

Endometriosis is a chronic, potentially disabling disease characterized by a proinflammatory environment, where dysbiosis in the oral–gut microbial community appears to play a prevalent role. Our findings reveal distinctive characteristics in the fecal microbiota composition of patients with endometriosis with an enrichment of *Facklamia* and *Corynebacterium* genera compared to control women, whose microbiota is enriched in *Alloprevotella*, *Succinivibrio*, and the *Rikenellaceae RC9* gut group. This compositional difference is accompanied by disease- and gut-specific functional enrichment in proinflammatory pathways previously linked to endometriosis development, paving the way for future studies exploring the potential mechanistic role of the oral–gut axis in disease pathogenesis.

## Figures and Tables

**Figure 1 medicina-61-00888-f001:**
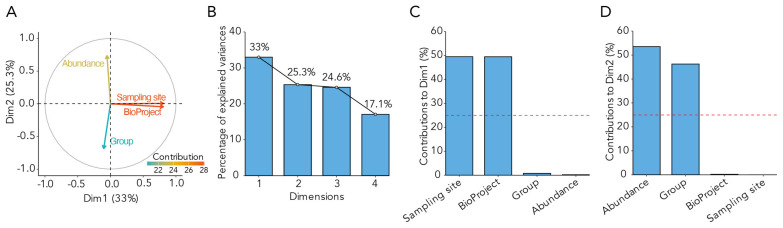
Principal component analysis (PCA) identifies major sources of variation across all samples. (**A**) Biplot displaying variable contributions to sample variance. Dim1 (33%) is strongly influenced by sampling site and BioProject (red and orange arrows), reflecting technical and cohort-related variability. Dim2 (25.3%) is more influenced by group (endometriosis status) and microbial abundance, suggesting a biological signal partially independent of technical confounding. (**B**) Bar plot showing variance explained by the first four principal components. (**C**,**D**) Contribution of individual variables to Dim1 and Dim2. The red dashed line represents the expected average contribution assuming equal influence across variables. These results highlight the need to account for technical confounders in meta-analyses; however, the orthogonal contribution of Group to Dim2 supports the presence of an underlying disease signal beyond cohort effects.

**Figure 2 medicina-61-00888-f002:**
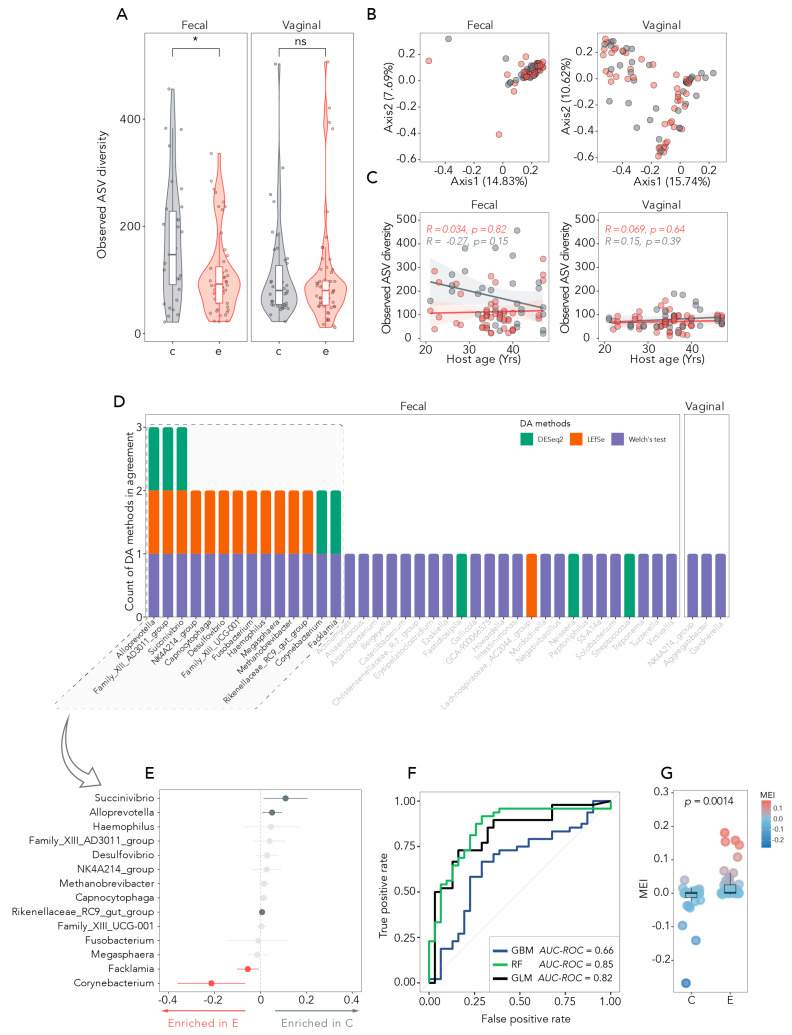
Microbiome composition and diversity analysis. (**A**) Violin plots comparing the observed alpha diversity (ASV richness) of fecal and vaginal microbiomes across different groups (**B**) Scatter plots illustrating the relationship between observed ASV diversity and host age for fecal and vaginal microbiomes. (**C**) Correlation analysis between observed ASV diversity and host age in fecal and vaginal microbiomes, including Pearson correlation coefficient (R) and *p*-value. (**D**) Bar plots showing the number of DA methods (DESeq2, LEFSe, Welch’s test) that agree on the enrichment of specific microbial taxa in fecal and vaginal microbiomes. (**E**) Dot plots depicting the relative abundance of selected microbial taxa in fecal and vaginal microbiomes, with taxa enriched in group endometriosis and group controls highlighted. (**F**) Receiver operating characteristic (ROC) curves evaluating the performance of machine learning models (GBM, RF, GLM) in differentiating fecal and vaginal microbiomes based on their microbial composition. (**G**) Bar plot comparing the mean importance of microbial taxa as predictors in the machine learning models. Red dots indicate woman with endometriosis, while grey dots indicate controls. Significance level: * *p* < 0.05, ns: Not Significant.

**Figure 3 medicina-61-00888-f003:**
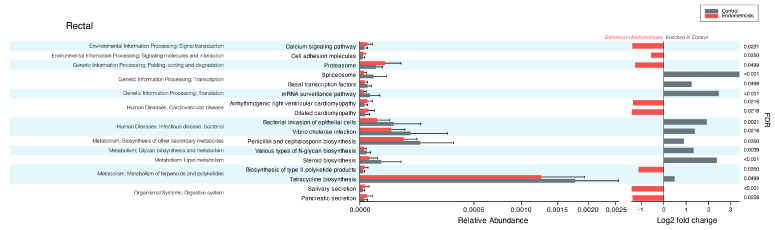
Functional microbial pathways differentially enriched in the intestinal environment of women with and without endometriosis. FDR: false discovery rate. See methods for details.

**Table 1 medicina-61-00888-t001:** Characteristics of women with and without endometriosis included in the meta-analysis.

Variable	Level	Overall	Control	Endometriosis	*p*-Value
		182	74	108	
Age (mean (SD))	-	35.5 (6.8)	36.6 (7.2)	34.9 (6.4)	0.117
BMI (mean (SD))	-	24.4 (3.7)	24.0 (2.5)	24.7 (4.4)	0.249
Sampling site (N (%))	Rectal	79 (43.4)	31 (41.9)	48 (44.4)	0.850
	Vaginal	103 (56.6)	43 (58.1)	60 (55.6)	
Menstrual Cycle Stage (N (%))	Follicular	86 (53.4%)	33 (50.8%)	53 (55.2%)	0.580
	Menses	75 (46.6%)	32 (49.2%)	43 (44.8%)	

## Data Availability

The datasets used during the current study are available from the corresponding authors on reasonable requests.

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
