# Peer review of "Meta-Analysis of 16S rRNA Sequencing Reveals Altered Fecal but Not Vaginal Microbial Composition and Function in Women with Endometriosis"

_medicina, 2025, doi:10.3390/medicina61050888_

Round 1
Reviewer 1 Report
Comments and Suggestions for Authors
The manuscript titled "Meta-analysis of 16S rRNA sequencing reveals altered fecal but not vaginal microbial composition and function in women with endometriosis" investigates the role of gut and vaginal microbiota in endometriosis. The study employs a meta-analytic approach to analyze 182 metagenomic sequences, identifying distinctive taxonomic and functional differences in the fecal microbiota of women with endometriosis compared to controls. The findings suggest a potential role for the oral-gut axis in endometriosis pathogenesis, supported by the development of a Microbial Endometriosis Index (MEI) and functional pathway analysis. The study is well-structured and addresses a relevant gap in the field, with clear implications for future research.
The manuscript is clear and well-organized, with a logical flow from introduction to discussion. The research question is highly relevant to the field of endometriosis and microbiome studies. The inclusion of both taxonomic and functional analyses strengthens the study's contributions. The meta-analytic approach is appropriate, but the exclusion of some studies due to lack of metadata (e.g., disease duration, antibiotic use) may limit the generalizability of the findings. The authors acknowledge this limitation, but further discussion on how these factors might influence the results would be beneficial. The use of multiple differential abundance analysis methods (Welch’s t-test, DESeq2, LEfSe) enhances robustness, but the rationale for selecting DESeq2 as the most conservative method for functional analysis could be elaborated. The methods section is detailed, but additional information on the parameters used for machine learning models (e.g., Random Forest, Gradient Boosting) would improve reproducibility. The data availability statement is adequate, but providing accession numbers for the included studies in the main text would enhance transparency. The conclusions are consistent with the evidence presented. The discussion effectively links the findings to existing literature and highlights the potential diagnostic and therapeutic implications of gut microbiota in endometriosis.
In addition, I would like to add some specific recomendations:
Line 235-240 (PCA Analysis): The PCA results are described, but the interpretation of how sampling site and BioProject contribute to variance could be expanded. For instance, how might these confounders impact the observed microbial differences?
Line 267-279 (Figure 2) The description of Figure 2 is thorough, but the figure itself could be improved by adding a key or legend to clarify the meaning of symbols/colors used in panels D-G.
Line 300-310 (Functional Pathways): the functional pathways identified are intriguing, but the biological relevance of pathways like "cardiomyopathy" in endometriosis is not fully explored. A deeper discussion on potential mechanistic links would strengthen this section.
Line 420-430 (Limitations): The limitations section is comprehensive, but the authors could briefly suggest how future studies might address these gaps (e.g., prospective designs with standardized metadata collection).
The manuscript presents a valuable contribution to understanding the role of gut microbiota in endometriosis. The findings are supported by robust methods and analyses, though minor improvements in clarity and discussion of limitations would enhance the work. With these revisions, the manuscript would be suitable for publication.
Author Response
REVIEWER 1
The manuscript is clear and well-organized, with a logical flow from introduction to discussion. The research question is highly relevant to the field of endometriosis and microbiome studies. The inclusion of both taxonomic and functional analyses strengthens the study's contributions. The meta-analytic approach is appropriate, but the exclusion of some studies due to lack of metadata (e.g., disease duration, antibiotic use) may limit the generalizability of the findings. The authors acknowledge this limitation, but further discussion on how these factors might influence the results would be beneficial.
RESPONSE
We sincerely thank the reviewer for this constructive feedback and the recognition of our study’s organization and methodological rigor.
We acknowledge that metadata heterogeneity and incompleteness—particularly regarding factors such as antibiotic usage, disease duration, hormonal therapy, and menstrual cycle phase—may confound microbial composition in meta-analytic studies. This is a well-recognized challenge in microbiome research and can indeed affect generalizability (10.1038/s41586-020-2881-9, 10.1038/s41467-017-01973-8). Several high-impact studies have demonstrated that antibiotic exposure can induce long-lasting shifts in gut microbiota composition, which may mask or mimic disease-specific signatures (10.1038/s41564-018-0257-9, 10.1038/s41586-022-04687-w). For instance, Palleja et al. in Nature Microbiology (2018) showed that a single antibiotic course can reduce microbiota diversity and induce compositional alterations lasting up to a year (10.1038/s41564-018-0257-9). Similarly, Zhang et al. (Nature, 2022) highlighted that even short-term antibiotic exposure could bias microbial function towards dysbiosis-associated pathways (10.1038/s41586-022-04687-w). Moreover, disease duration is another variable with important implications, as chronicity may affect immune–microbiota interactions. Chronic inflammatory environments have been shown to perpetuate dysbiotic feedback loops (10.1016/j.cell.2014.03.011). However, the majority of public datasets, especially vaginal ones, lack granular metadata on disease progression or patient treatment history, limiting stratified subgroup analyses. This limitation is inherent to current metagenomic repositories and has been consistently cited in large-scale microbiome meta-analyses (10.1038/s41591-019-0406-6, 10.1172/jci.insight.171311).
So, we have now updated the manuscript at pages 12-13, lines 522-534, as follow:
While the exclusion of datasets lacking metadata (e.g., disease duration, antibiotic exposure, gastrointestinal symptomatology) may influence the generalizability of our results, such limitations are common in large-scale microbiome meta-analyses. Prior studies have demonstrated that antibiotic usage and chronic inflammation significantly modulate gut microbial composition (10.1038/s41564-018-0257-9; 10.1038/s41586-022-04687-w). Moreover, the lack of detailed clinical information, such as gastrointestinal symptom profiles, limited the possibility of correlating microbial alterations with symptomatology directly related to the hypothesized pathophysiological mechanisms. Metadata completeness remains inconsistent across repositories, constraining stratified subgroup analysis. Future prospective studies with harmonized clinical and environmental metadata are warranted to validate and expand these findings.
The use of multiple differential abundance analysis methods (Welch’s t-test, DESeq2, LEfSe) enhances robustness, but the rationale for selecting DESeq2 as the most conservative method for functional analysis could be elaborated.
RESPONSE
We agree that elaborating on the rationale for choosing DESeq2 as the primary method for functional differential abundance analysis could improve transparency. Among the methods tested, DESeq2 provided the most conservative and reproducible estimates, particularly in sparsely distributed functional gene counts. Its statistical framework, originally developed for RNA-seq, incorporates shrinkage estimation for dispersion and fold change, which reduces false positives - an issue known in metagenomic data due to zero inflation and high inter-sample variability (10.1186/s13059-014-0550-8). Moreover, DESeq2 has been extensively validated in microbiome studies for its ability to control type I error under conditions of uneven library size and heteroscedasticity (10.1371/journal.pcbi.1003531, 10.1186/s40168-017-0237-y). While LEfSe excels in biomarker discovery due to its effect-size emphasis, it can produce more liberal p-values in compositional data. In contrast, DESeq2’s conservative nature aligns better with our meta-analytic design, where reducing false discovery across heterogeneous datasets is a priority.
So, the manuscript was updated accordingly at pages 5 (lines 239-243):
DESeq2 was selected as the primary tool for functional inference due to its robustness in controlling type I error in sparse and overdispersed metagenomic data. Compared to LEfSe and t-test-based methods, DESeq2 applies variance-stabilizing transformations and shrinkage estimators that enhance reproducibility, particularly when working with uneven sampling depth across public datasets (10.1371/journal.pcbi.1003531, 10.1186/s40168-017-0237-y).
The methods section is detailed, but additional information on the parameters used for machine learning models (e.g., Random Forest, Gradient Boosting) would improve reproducibility.
RESPONSE
We thank the reviewer for this comment. To enhance reproducibility, we have now included a more detailed description of the hyperparameters and modeling strategy for the machine learning analyses. We confirm that we employed the caret package (v6.0-94) in R and trained models using repeated 10-fold cross-validation (3 repeats) with AUC as the optimization metric. The models were trained on a balanced set of microbial genera and host features (including age), after excluding missing values. Specifically:
Gradient Boosting Machine (GBM):
The model was trained using the gbm method in caret, with tuning performed over a default grid of shrinkage (0.1), interaction depth (1–5), and number of trees (50–150). The optimal model had interaction.depth = 3, n.trees = 100, and shrinkage = 0.1.
Random Forest (RF):
Trained using the rf method with 500 trees (ntree = 500). The mtry parameter was optimized within cross-validation and selected automatically by caret. The final model used mtry = 2.
Generalized Linear Model (GLM):
A binomial logistic regression (family = binomial) was fit to the same training data using MEI and other selected features. No regularization was applied to preserve interpretability.
All models were evaluated using the MLeval package for ROC, AUC, and precision-recall performance metrics. We now report these methodological details in the revised Methods.
Added to manuscript (Methods section page 5, lines 214-221):
Machine learning classifiers were trained using the caret R package (v6.0-94). Ten-fold cross-validation repeated three times was used for model validation (trainControl(method = "repeatedcv", number = 10, repeats = 3)). GBM models were optimized over shrinkage values of 0.1 and interaction depths of 1–5; the final model had 100 trees and depth = 3. RF models were trained with 500 trees, with mtry optimized within cross-validation. Logistic regression models used MEI and other features without regularization. All models were evaluated using the MLeval package to compute ROC curves, AUC, and classification metrics.
The data availability statement is adequate, but providing accession numbers for the included studies in the main text would enhance transparency.
RESPONSE
We thank the Reviewer for the suggestion. The BioProject accession numbers (PRJNA722289 and PRJNA424567) corresponding to the included studies are already provided in the manuscript (Material and Methods, page 3, lines 112–113). We have carefully checked and ensured that they are clearly reported to enhance transparency and facilitate data accessibility.
The conclusions are consistent with the evidence presented. The discussion effectively links the findings to existing literature and highlights the potential diagnostic and therapeutic implications of gut microbiota in endometriosis.
In addition, I would like to add some specific recomendations:
Line 235-240 (PCA Analysis): The PCA results are described, but the interpretation of how sampling site and BioProject contribute to variance could be expanded. For instance, how might these confounders impact the observed microbial differences?
RESPONSE
We thank the reviewer for highlighting this important point. As shown in Figure 1, sampling site and BioProject contribute substantially to Dim1, which captures ~33% of total variance. These variables represent known technical confounders in microbiome meta-analyses and likely reflect differences in sequencing platforms, sample processing protocols, and DNA extraction methods, which can systematically shift community profiles independent of biological status. Importantly, while Dim1 is largely driven by these technical factors, our disease signal (i.e., Group, indicating endometriosis status) contributes more to Dim2 - a component partially orthogonal to Dim1. This separation suggests that disease-associated microbial signatures are not fully confounded by sampling site or BioProject effects. Additionally, we conducted downstream analyses (e.g., differential abundance, machine learning) within biome-specific subsets (fecal and vaginal), which further mitigates the impact of inter-biome and inter-cohort variability. To address the reviewer’s point more explicitly, we have expanded the figure legend and provided a new paragraph in the Results section (pages 6-7, lines 272-280) interpreting how confounding by technical factors was accounted for.
Line 300-310 (Functional Pathways): the functional pathways identified are intriguing, but the biological relevance of pathways like "cardiomyopathy" in endometriosis is not fully explored. A deeper discussion on potential mechanistic links would strengthen this section.
RESPONSE
We thank the reviewer for this valuable comment. As suggested, we have expanded the discussion regarding the biological relevance of cardiomyopathy-related pathways identified in our analysis. In particular, we highlighted that microbiota dysbiosis, which has been consistently associated with chronic inflammatory states such as atherosclerosis, may also play a role in linking endometriosis to cardiovascular disease. Furthermore, recent evidence suggests an increased risk of heart failure among women with endometriosis, with heart failure often representing the clinical manifestation of underlying cardiomyopathic processes. We also discussed potential mechanistic links, including systemic inflammation, oxidative stress, endothelial dysfunction, and hormonal imbalances (particularly involving estrogen regulation), which are known contributors to both endometriosis and cardiovascular diseases. Therefore, the enrichment of cardiomyopathy-associated pathways in the fecal microbiome of women with endometriosis could reflect shared pathogenic mechanisms, reinforcing the systemic inflammatory and metabolic components of the disease.
We have revised the manuscript at page 11, lines 446-453) accordingly to better address this important point, as follow:
Additionally, recent studies have reported an increased risk of heart failure among women with endometriosis, a condition that often represents the clinical manifestation of various forms of cardiomyopathy [10.1136/heartjnl-2024-324675]. Mechanisms such as systemic inflammation, oxidative stress, endothelial dysfunction, and imbalances in estrogenic hormonal regulation are likely to underlie this association[10.1097/OGX.0000000000000718 ]. In this context, the identification of pathways related to cardiomyopathy in our study may reflect the presence of shared pathogenic processes between endometriosis and cardiovascular diseases.
Line 420-430 (Limitations): The limitations section is comprehensive, but the authors could briefly suggest how future studies might address these gaps (e.g., prospective designs with standardized metadata collection).
RESPONSE
We thank the reviewer for this constructive and important suggestion.
We fully agree that outlining possible strategies for future research enhances the utility of the limitations section.
In response, we have added the following sentence at the end of the limitations paragraph (page 13, lines 530-531), as follow:
Future prospective studies with harmonized clinical and environmental metadata are warranted to validate and expand these findings.
We believe that the adoption of prospective study designs, together with standardized and comprehensive metadata collection, will not only improve reproducibility but also facilitate integrative analyses across different cohorts.
This approach could ultimately lead to a more precise understanding of the complex relationships described in the current study.
The manuscript presents a valuable contribution to understanding the role of gut microbiota in endometriosis. The findings are supported by robust methods and analyses, though minor improvements in clarity and discussion of limitations would enhance the work. With these revisions, the manuscript would be suitable for publication.
Reviewer 2 Report
Comments and Suggestions for Authors
This is an interesting study on the association of gut and vaginal microbiome in the pathophysiology of endometriosis. I have a few minor comments to make:
1) Line 53: In my experience, most intestinal symptoms severe enough to be noticed end up being caused by endometriosis loci and or adhesions directly on the intestinal tissues, causing mechanical obstruction or inflammation. Please specify what types of symptoms according to the literature are attributed to the immune-dysregulation observed in endometriosis.
2) Lines 82-93: Is this paragraph left in the manuscript by mistake? It seems disconnected to the rest of the text. This section seems like instructions taken directly from a checklist or database. Please remove or modify.
3) Lines 133-134: Please provide citations for the listed indices.
4) Lines 232-233: Where there any differences with regard to intestinal symptoms? Gastrointestinal symptomatology was at the core of the pathogenesis theory you presented in the introduction, therefore one would think that its inclusion would be important for the purposes of this study.
5) Discussion is well-written and the limitations of the work are discussed, no need for revisions.
Author Response
REVIEWER 2:
This is an interesting study on the association of gut and vaginal microbiome in the pathophysiology of endometriosis. I have a few minor comments to make:
1) Line 53: In my experience, most intestinal symptoms severe enough to be noticed end up being caused by endometriosis loci and or adhesions directly on the intestinal tissues, causing mechanical obstruction or inflammation. Please specify what types of symptoms according to the literature are attributed to the immune-dysregulation observed in endometriosis.
RESPONSE
We thank the Reviewer for the valuable comment. As requested, we have specified in the manuscript the types of gastrointestinal symptoms reported in the literature that are attributed to immune dysregulation in endometriosis. In particular, we now distinguish between symptoms caused by mechanical obstruction (e.g., bowel obstruction due to lesions) and functional symptoms (e.g., cyclic bloating, diarrhea, constipation) likely related to inflammatory and immune-mediated processes, independent of direct intestinal involvement. Relevant references have been added to support these statements (page 2, lines 54-59).
2) Lines 82-93: Is this paragraph left in the manuscript by mistake? It seems disconnected to the rest of the text. This section seems like instructions taken directly from a checklist or database. Please remove or modify.
RESPONSE
We thank the reviewer for the observation. The paragraph was inadvertently included in the manuscript during the editing process. We have now removed it to improve the clarity and coherence of the text.
3) Lines 133-134: Please provide citations for the listed indices.
RESPONSE
We thank the reviewer for the valuable observation. Upon careful revision of the manuscript, we realized that an error had occurred: we initially stated that the Shannon index was used, whereas in fact the Observed Alpha Diversity Index was employed in our analysis. We have corrected this mistake in the revised version of the manuscript. Furthermore, we have now added the appropriate reference supporting the use of the Observed Alpha Diversity Index and the Bray-Curtis similarity index. These citations have been incorporated into the main text and the reference list accordingly to improve the scientific rigor and completeness of the manuscript:
- Observed-alpha diversity index [https://doi.org/10.1038/s41598-024-77864-y]
- Bray-Curtis similarity index [Bray, J. Roger, and John T. Curtis. "An ordination of the upland forest communities of southern Wisconsin." Ecological monographs4 (1957): 326-349.]
4) Lines 232-233: Where there any differences with regard to intestinal symptoms? Gastrointestinal symptomatology was at the core of the pathogenesis theory you presented in the introduction, therefore one would think that its inclusion would be important for the purposes of this study.
RESPONSE
We sincerely thank the reviewer for this insightful comment. As previously clarified in our response to Reviewer 1, the metadata associated with the publicly available sequencing datasets were not sufficiently granular. In particular, information regarding gastrointestinal symptomatology was not available. The lack of such detailed clinical data limited our ability to directly correlate microbial alterations with intestinal symptoms. We agree that the inclusion of these data would have been highly valuable for further strengthening the pathophysiological link proposed in the introduction. This limitation has been acknowledged and discussed in the revised version of the manuscript (pages 12-13, lines 522-534).
5) Discussion is well-written and the limitations of the work are discussed, no need for revisions.
Reviewer 3 Report
Comments and Suggestions for Authors
This study aimed to assess the microbial landscape and pathogenic potential of distinct biological niches during endometriosis. The findings that altered fecal but not vaginal microbial composition and function are associated with endometriosis are very interesting. The Manuscript is soundly written, the clinical question and available literature are properly explained, and the data are informative. Still, some aspects of the data presentation need revision and better explanation to improve the clarity and justify the overall conclusions.
The third and the fourth paragraph of Methods section are redundant and should be deleted or completely rewriten to clearly present the methodology of this study.
Authors should explain with more details how was the control group formed. Were the control patients also taken from bioprojects (PRJNA722289 106 and PRJNA424567) or somewhere else? What were the criteria for inclusion in the control group? Was the control group matched in any way with the study group?
Comments on the Quality of English LanguageThe English language is generally good, but some sentences are too long and hard to follow. English editing is suggested.
Author Response
REVIEWER 3
This study aimed to assess the microbial landscape and pathogenic potential of distinct biological niches during endometriosis. The findings that altered fecal but not vaginal microbial composition and function are associated with endometriosis are very interesting. The Manuscript is soundly written, the clinical question and available literature are properly explained, and the data are informative. Still, some aspects of the data presentation need revision and better explanation to improve the clarity and justify the overall conclusions.
The third and the fourth paragraph of Methods section are redundant and should be deleted or completely rewriten to clearly present the methodology of this study.
RESPONSE
Thank you for your thoughtful and constructive comments regarding the Methods section of our manuscript.
We have carefully revised the third and fourth paragraphs to improve clarity and eliminate redundancy. The updated section now presents a clearer description of the study selection process and methodological approach.
Authors should explain with more details how was the control group formed. Were the control patients also taken from bioprojects (PRJNA722289 106 and PRJNA424567) or somewhere else? What were the criteria for inclusion in the control group? Was the control group matched in any way with the study group?
RESPONSE
We thank the reviewer for the valuable observation In particular, we addressed your specific questions as follows:
- Formation of the control group: We clarified that the control individuals were selected from the same two bioprojects (PRJNA722289 and PRJNA424567) as the cases, and were labeled as “healthy” or “control” by the original authors.
- Inclusion criteria for controls: We now explicitly state that inclusion required the availability of raw 16S rRNA sequencing data, relevant metadata, and designation as “control” in the source studies.
- Matching: We added a description of how the control group was frequency-matched to the endometriosis group based on sample type (fecal or vaginal) and, when possible, age distribution.
All these clarification are now in the Materials and Methods section of the revised manuscript (page 3, lines 119-130).